# Forb diversity globally is harmed by nutrient enrichment but can be rescued by large mammalian herbivory
Rebecca A. Nelson [1,2] ✉, Lauren L. Sullivan[3,4,5], Erika I. Hersch-Green [6], Eric W. Seabloom [7], Elizabeth T. Borer [7], Pedro M. Tognetti [8], Peter B. Adler [2], Lori Biederman [9], Miguel N. Bugalho [10], Maria C. Caldeira [11], Juan P. Cancela [12], Luísa G. Carvalheiro [13], Jane A. Catford [14,15], Chris R. Dickman[16], Aleksandra J. Dolezal [17], Ian Donohue [18], Anne Ebeling [19], Nico Eisenhauer [20,21], Kenneth J. Elgersma [22], Anu Eskelinen [23], Catalina Estrada [24], Magda Garbowski[25], Pamela Graff [8,26], Daniel S. Gruner [27], Nicole Hagenah[28], Sylvia Haider[29], W. Stanley Harpole [20,30,31], Yann Hautier [32], Anke Jentsch [33], Nicolina Johanson [23], Sally E. Koerner [34], Lucíola S. Lannes[35], Andrew S. MacDougall [17], Holly Martinson [36], John W. Morgan[37], Harry Olde Venterink [38], Devyn Orr[39], Brooke B. Osborne[40], Pablo L. Peri [41], Sally A. Power [42], Xavier Raynaud[43], Anita C. Risch [44], Mani Shrestha [33,45], Nicholas G. Smith [46], Carly J. Stevens [47], G. F. Ciska Veen [48], Risto Virtanen [23], Glenda M. Wardle [16], Amelia A. Wolf [49], Alyssa L. Young [34] & Susan P. Harrison[1]

Forbs ("wildflowers") are important contributors to grassland biodiversity but are vulnerable to environmental changes. In a factorial experiment at 94 sites on 6 continents, we test the global generality of several broad predictions: (1) Forb cover and richness decline under nutrient enrichment, particularly nitrogen enrichment. (2) Forb cover and richness increase under herbivory by large mammals. (3) Forb richness and cover are less affected by nutrient enrichment and herbivory in more arid climates, because water limitation reduces the impacts of competition with grasses. (4) Forb families will respond differently to nutrient enrichment and mammalian herbivory due to differences in nutrient requirements. We find strong evidence for the first, partial support for the second, no support for the third, and support for the fourth prediction. Our results underscore that anthropogenic nitrogen addition is a major threat to grassland forbs, but grazing under high herbivore intensity can offset these nutrient effects.

Forbs, or non-graminoid herbaceous angiosperms[1], play essential roles in maintaining grassland diversity, structure, and function through supporting pollinator populations[2–5]. Anthropogenic global change drivers, however, have altered forb-rich communities, such as grasslands, in ways not yet fully understood. Most field studies to date have been performed at the level of an individual site or single stressor, even though grassland forbs are widespread, diverse, and threatened by multiple, interacting anthropogenic drivers[6–9]. Nutrient enrichment[10,11], changes in large mammalian herbivores through the loss of native megafauna or the addition of livestock[10,12,13], and climate change[14] are major contributors to losses in forb biodiversity. Based on predictions from coexistence theory[15–18], fertilization[19–23] and the loss of large mammalian herbivores[24–28] can decrease the diversity and abundance

of shorter, slower-growing forbs through increasing competition from taller, faster-growing grasses[29–32] for light[33,34], even for initially dominant species[35]. Aridity, which tends to decrease forb richness and abundance[14], may dampen these fertilization and herbivore effects as water limitation reduces the impacts on forbs from competition with grasses[36–38]. However, empirical support from single-site studies for the importance of grass-forb competition as a mediator of global change effects on forb biodiversity remains equivocal due to the contingencies that arise with biogeographic, ecological, and climatic variation[5,39–43]. Fertilization effects may depend upon which type and in which combinations nutrients are added. Nitrogen enrichment, in particular, favors grasses at the expense of forbs[10] due to nitrogen's role as a more prevalent limiting nutrient than potassium or phosphorus[44–46].

Studying the combined effects of nutrient enrichment, changes in large herbivores, and aridity gradients as interacting drivers of global change rather than isolating them fills a critical knowledge gap[9,39] and more accurately reflects real-world scenarios of anthropogenic change[6–8].

In a factorial experiment at 94 grassland sites on 6 continents, we tested the global generality of several broad predictions arising from previous studies. We thus predict that (1) Forb cover and richness will decline under nutrient enrichment, particularly nitrogen enrichment, benefitting grasses at the expense of forbs. (2) Forb cover and richness will increase under herbivory by large mammals, especially when nutrients are enriched, as grazing will offset the effects of increased grass competition on forbs under fertilization. (3) Forb richness and cover will be less affected by nutrient enrichment and herbivory in more arid climates. (4) Different forb families will respond differently to nutrient enrichment and mammalian herbivory due to differences in nutrient requirements and tolerances. We found strong evidence for the first, partial support for the second, no support for the third prediction, and strong evidence for the fourth prediction. Forb richness and cover are reduced by nutrient addition, with nitrogen having the greatest effect; forb cover is enhanced by large mammal herbivory, although only under conditions of nutrient enrichment and high herbivore intensity; and forb richness is lower in more arid sites regardless of nutrient level or the presence of herbivores. We also found that nitrogen enrichment disproportionately affects forbs in certain families including Asteraceae and Fabaceae, two large families that are essential for pollination, biomass production, and nutrient cycling[5,10]. In contrast, Gerianaceae and Apiaeae did not respond to nitrogen enrichment, while nitrogen enrichment increased Polygonaceae cover. Our results underscore that eutrophication, especially nitrogen addition, is a major threat to grassland forbs and the ecosystem services they support, but large mammalian herbivory can offset these effects.

## Results
### Nutrient effects
In support of our first prediction, NPKμ fertilization contributed to forb declines (Fig. 1, Table 1). Fertilization via combined nitrogen, phosphorus, potassium with micronutrient enrichment (NPKμ treatment) decreased forb species richness by 27% (see "Methods") ($t = -8.11$, $p < 0.001$), forb family richness by 19% ($t = -6.04$, $p < 0.001$), and forb cover by 13% ($t = -2.07$, $p = 0.038$) (Table 1). NPKμ fertilization also decreased grass species richness by 8%, ($t = -2.50$, $p = 0.012$) but increased total grass cover by 22% ($t = 3.27$, $p = 0.001$).

As predicted, nitrogen was the strongest contributor to declines in forb richness and cover compared to phosphorus and potassium with micronutrients (Fig. 2, Table 2). Forb species richness decreased in response to nitrogen by 14% ($t = -6.12$, $p < 0.001$), to phosphorus by 6% ($t = -2.32$, $p = 0.020$), and potassium and micronutrients by 6% ($t = -2.59$, $p = 0.010$)

with no interactions among nutrients (Table 2). Likewise, forb family richness and forb cover in the fertilization factorial experiment decreased in response to nitrogen by 7% and 7%, respectively (family richness: $t = -3.49$, $p < 0.001$; forb cover: $t = -3.18$, $p = 0.001$), but did not respond to phosphorus addition nor potassium and micronutrients (Table 2, Tables S1-S3).

Responses of common forb families further supported this trend and our prediction that fertilization effects vary by forb family. Asteraceae species richness decreased more in response to nitrogen than phosphorous and potassium with micronutrients (Fig. 3, Table S4), and Fabaceae cover decreased in response to nitrogen while increasing in response to phosphorus and potassium with micronutrients (Fig. 3, Tables S4−S6) However, nitrogen enrichment did not affect Geraniaceae species richness, which instead declined under phosphorus enrichment ($t = -3.17$, $p = 0.002$), nor Apiaceae cover and richness (Table S4). In contrast, Polygonaceae cover increased with nitrogen enrichment ($t = 3.62$, $p < 0.001$) but declined with added potassium with micronutrients (Table S4). Grass cover increased with nitrogen addition ($t = 4.44$, $p < 0.001$) and phosphorus addition. ($t = 5.29$, $p < 0.001$).

### Herbivore effects
In partial support of our second prediction, large mammalian herbivore exclusion under fertilization and high herbivore intensity, as measured by the difference in live biomass between the control and the fenced treatments, contributed to forb declines (Fig. 1, Table 1, Tables S7-S9). Herbivore exclusion via fencing did not directly affect forb or grass richness and cover estimates (Fig. 1, Table 1). High herbivore intensity, however, alleviated the suppression of forb species richness by NPKμ fertilization ($t = -1.96$, $p = 0.0499$) such that forb richness no longer was negatively affected by NPKμ when herbivore intensity was high. For sites with high herbivore intensity, herbivory alleviated the suppression of forb cover under fertilization, although forb cover was not strongly affected by the exclusion of herbivores under ambient conditions ($t = 3.06$, $p = 0.002$) (Table 1). Herbivore effects further varied considerably among common forb families (Figs. S1-S5, Table S9). Asteraceae richness and cover were highest at sites with high herbivore intensity but were strongly suppressed under fertilized conditions when herbivores were removed from the high-intensity sites (Fig. S2, Tables S9-S10). In contrast, Fabaceae and Apiaceae richness and cover were not affected by herbivore exclusion nor intensity (Tables S9-S10). Geraniaceae cover was enhanced by the interaction between herbivore exclusion and fertilization, while Polygonaceae richness was positively associated with herbivore intensity but dampened by interactions between fertilization and fencing (Tables S9-S10).

### Aridity effects
Contrary to our third prediction, potential evapotranspiration (PET), a measure of aridity, did not interact with herbivore exclusion and nutrient

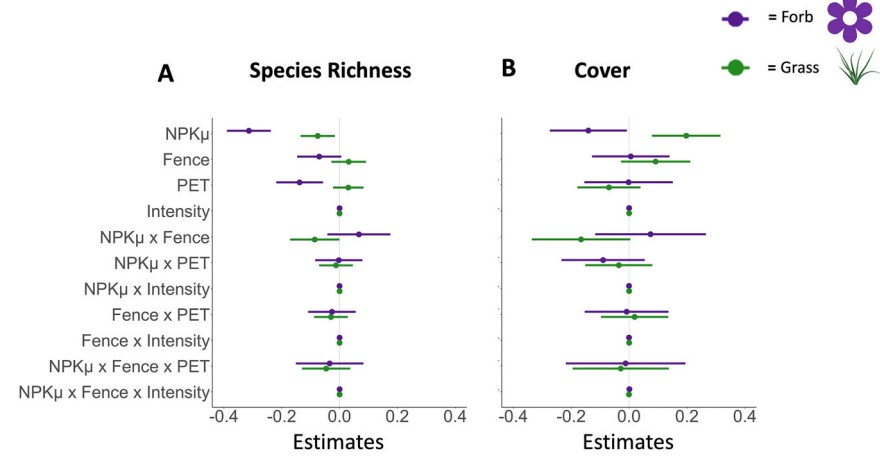

**Fig. 1 | Fencing by fertilization effects.** The effect of the herbivore exclusion via fencing treatment, fertilization treatment (fencing by fertilization experiment), potential evapotranspiration (PET), and herbivore intensity on (**A**) species richness and (**B**) cover for forbs (purple) and grasses (green). Model estimates of log response ratios for the effect of different treatments are shown relative to the control treatment (estimate = 0). Binary response variables were converted to log response variables to account for the change from pre-treatment to current data and cover data were normalized relative to maximum plot cover. Fence refers to herbivore exclusion fencing. NPKμ refers to the nitrogen, phosphorus, and potassium with micronutrient treatment. Error bars show the 89% confidence interval. BioRender. (2025). *Image icon of green grass*. BioRender. Retrieved from https://biorender.com. *N* = 82 sites.

## Table 1 | Fencing by fertilization results

| Predictors | Forb Species Richness | | Forb Family Richness | | Forb Cover | | Grass Species Richness | | Grass Cover | |
|---|---|---|---|---|---|---|---|---|---|---|
| | Estimates | t-value; p-value | Estimates | t-value; p-value | Estimates | t-value; p-value | Estimates | t-value; p-value | Estimates | t-value; p-value |
| Intercept | −0.00 (−0.11 to 0.10) | −0.08; 0.933 | −0.02 (−0.10 to 0.07) | −0.39; 0.699 | 0.09 (−0.14 to 0.31) | 0.75; 0.454 | −0.04 (−0.11 to 0.03) | −1.18; 0.239 | −0.11 (−0.26 to 0.04) | −1.49; 0.136 |
| NPKμ | −0.31 *** (−0.39 to −0.24) | **−8.11; <0.001** | −0.21 *** (−0.28 to −0.14) | **−6.04; <0.001** | −0.14 * (−0.27 to −0.01) | **−2.07; 0.038** | −0.08 * (−0.13 to −0.02) | **−2.50; 0.012** | 0.20 ** (0.08 to 0.32) | **3.27; 0.001** |
| Fence | −0.07 (−0.15 to 0.01) | −1.80; 0.072 | 0.01 (−0.06 to 0.07) | 0.16; 0.870 | 0.01 (−0.13 to 0.14) | 0.09; 0.932 | 0.03 (−0.03 to 0.09) | 1.04; 0.300 | 0.09 (−0.03 to 0.21) | 1.50; 0.133 |
| PET | −0.14 *** (−0.22 to −0.06) | **−3.36; 0.001** | −0.14 *** (−0.21 to −0.07) | **−3.88; <0.001** | −0.00 (−0.15 to 0.15) | −0.02; 0.982 | 0.03 (−0.02 to 0.08) | 1.13; 0.259 | −0.07 (−0.18 to 0.04) | −1.25; 0.210 |
| Intensity | 0.00 (−0.00 to 0.00) | 0.07; 0.945 | 0.00 (−0.00 to 0.00) | 0.14; 0.892 | 0.00 (−0.00 to 0.00) | 0.30; 0.762 | −0.00 * (−0.00 to −0.00) | **−2.60; 0.013** | 0.00 (−0.00 to 0.00) | 0.19; 0.854 |
| NPKμ × Fence | 0.07 (−0.04 to 0.18) | 1.21; 0.228 | 0.04 (−0.06 to 0.14) | 0.80; 0.425 | 0.07 (−0.12 to 0.27) | 0.76; 0.447 | −0.09 * (−0.17 to −0.00) | **−1.98; 0.048** | −0.17 (−0.34 to 0.00) | −1.91; 0.056 |
| NPKμ × PET | −0.00 (−0.08 to 0.08) | −0.06; 0.949 | 0.01 (−0.07 to 0.08) | 0.17; 0.865 | −0.09 (−0.23 to 0.05) | −1.22; 0.221 | −0.01 (−0.07 to 0.05) | −0.42; 0.676 | −0.04 (−0.15 to 0.08) | −0.60; 0.549 |
| NPKμ × Intensity | −0.00 * (−0.00 to −0.00) | **−1.96; 0.050** | −0.00 (−0.00 to 0.00) | −1.03; 0.305 | −0.00 (−0.00 to 0.00) | −1.74; 0.082 | 0.00 (−0.00 to 0.00) | 0.30; 0.767 | 0.00 (−0.00 to 0.00) | 0.15; 0.879 |
| Fence × PET | −0.03 (−0.11 to 0.06) | −0.63; 0.528 | −0.03 (−0.10 to 0.05) | −0.68; 0.494 | −0.01 (−0.15 to 0.14) | −0.12; 0.907 | −0.03 (−0.09 to 0.03) | −1.00; 0.315 | 0.02 (−0.10 to 0.14) | 0.32; 0.750 |
| Fence × Intensity | 0.00 (−0.00 to 0.00) | 0.11; 0.909 | 0.00 (−0.00 to 0.00) | 1.57; 0.116 | −0.00 (−0.00 to 0.00) | −1.13; 0.257 | 0.00 (−0.00 to 0.00) | 0.65; 0.514 | 0.00 (−0.00 to 0.00) | 1.14; 0.255 |
| NPKμ×Fence × PET | −0.03 (−0.15 to 0.08) | −0.57; 0.566 | 0.01 (−0.09 to 0.12) | 0.27; 0.787 | −0.01 (−0.22 to 0.19) | −0.11; 0.909 | −0.05 (−0.13 to 0.04) | −1.09; 0.277 | −0.03 (−0.20 to 0.14) | −0.34; 0.732 |
| NPKμ × Fence × Intensity | 0.00 (−0.00 to 0.00) | 0.95; 0.340 | −0.00 (−0.00 to 0.00) | 0.76; 0.449 | 0.00 ** (0.00 to 0.00) | **3.06; 0.002** | 0.00 (−0.00 to 0.00) | 0.23; 0.821 | −0.00 (−0.00 to 0.00) | −1.64; 0.101 |
| **Random Effects** | | | | | | | | | | |
| σ² | 0.28 | | 0.23 | | 0.91 | | 0.16 | | 0.64 | |
| τ00 | 0.30 block | | 0.24 block | | 0.67 block | | 0.17 block | | 0.41 block | |
| | 0.14 site_code | | 0.11 site_code | | 0.29 site_code | | 0.08 site_code | | 0.21 site_code | |
| N | 6 block | | 6 block | | 6 block | | 6 block | | 6 block | |
| | 46 site_code | | 46 site_code | | 46 site_code | | 48 site_code | | 48 site_code | |
| Observations | 3500 | | 3500 | | 3500 | | 3535 | | 3535 | |

Mixed effects model results for the effects of fertilization by fencing, herbivore intensity, and potential evapotranspiration (PET) on Forb Family Richness, Forb Species Richness, Forb Cover, Grass Species Richness, and Normalized Grass Cover. All response variables were calculated using LRRs. The intercept is the mean value of the unfenced and unfertilized control plots. The parenthetical numbers are the confidence interval. Bolded numbers indicate statistical significance. $N$ = 82 biologically independent sites.

$*p < 0.05$ $**p < 0.01$ $***p < 0.001$.

**Fig. 2 | Fertilization factorial effects.** The effect of different nutrients (fertilization factorial experiment) on (**A**) species richness and (**B**) cover for forbs (purple) and grasses (green). Model estimates for log response ratios are shown relative to the control treatment (estimate = 0). Response variables were converted to log response variables to account for the change from pretreatment to current data with cover data normalized by maximum plot cover. Multiple nutrient interactions are included for richness. *N* refers to nitrogen, *P* refers to phosphorus, and Kμ to potassium with micronutrients. Error bars show the 89% confidence interval. BioRender. (2025). *Image icon of green grass.* BioRender. Retrieved from https://biorender.com. *N* = 89 sites.

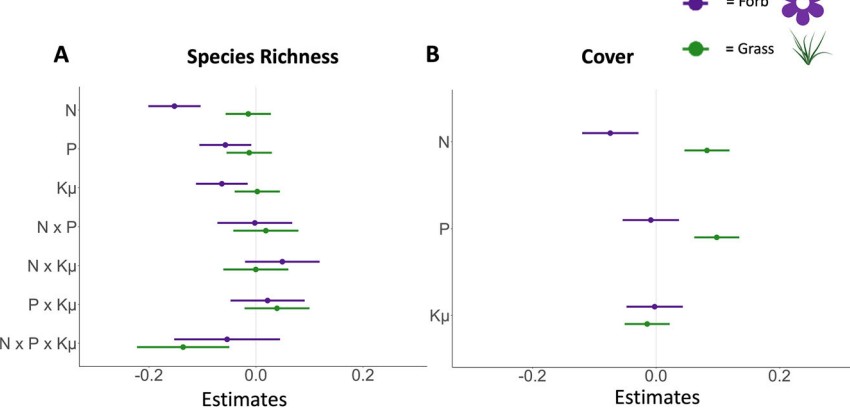

enrichment to modify forb or grass richness or cover (Table 1). Forb species and family richness both decreased with increasing PET (Fig. 1, Table 1). PET was negatively associated with forb family richness ($t = -3.88$, $p < 0.001$), forb species richness ($t = -3.36$, $p = 0.001$) but not with forb cover, grass richness, nor grass cover (Table 1).

## Discussion

Across diverse grasslands spanning a global range of climate conditions, nutrient enrichment, particularly nitrogen, reduced forb richness, while increasing grass cover. Herbivory by large mammals, however, can rescue forb diversity declines from these detrimental nutrient effects, especially at sites where herbivore intensity was high. Large mammalian herbivores at naturally occurring densities play an essential role in offsetting these negative effects of fertilization on forbs by decreasing competition from grasses through consuming grasses. Forb richness declined with increasing potential evapotranspiration (PET), but PET did not interact with herbivore exclusion and fertilization. Our findings highlight that fertilization, especially nitrogen enrichment, is a major threat to grassland forb diversity. Furthermore, nutrient enrichment especially threatened richness and cover of Asteraceae and Fabaceae, two of the largest plant families that are key providers of pollinator support and strongly contribute to food security, nutrient cycling, and productivity[5,10,47].

In support of our first prediction regarding nutrients, forb richness decreased under fertilization, while grass cover increased, producing functional shifts from a more diverse forb-dominated community to a less forb-rich, more grass-dominated system. These findings support past studies that have found fertilization increases grass dominance at the expense of forbs[10,20,22,33,46,48] including for initially dominant forb species[35]. Taller, faster-growing grasses favored under nutrient enrichment may shade out more light-demanding, shorter, slower-growing forbs[20,33], altering grass-forb community assembly and coexistence[17,38,49]. Losses in forb richness under fertilization, as in this study, suggest that extinctions of forb species, likely those of smaller stature with lower competitive ability for light[34,50], can drive the negative impact of nutrient enrichment on overall community diversity. Losses in forb diversity can have consequences for the provisioning of ecosystem services such as pollination, food security, and medicinal plants[5].

In further support of our first prediction, nitrogen addition, more than phosphorus or potassium with micronutrients, strongly contributed to declines in forb cover and forb richness, while increasing grass cover. This suggests that nitrogen addition may have more severe consequences for forb declines than the addition of other nutrients by giving grasses a competitive advantage[10,45,51]. Outcomes of grass versus forb competition may thus depend on type of nutrient[52]. In general, nitrogen enrichment is likely to have a pervasive impact on forbs because of its mobility and widespread inputs through airborne nutrient deposition[44,53]. While at a global scale, nitrogen is typically a growth-limiting nutrient, these effects can vary

regionally as under heavy nitrogen fertilization, other nutrients like phosphorus and potassium can become the limiting nutrients instead with nutrient colimitation more widespread than single-nutrient limitation[52,54].

In partial support of our second prediction, herbivory offset the negative effects of fertilization on forb cover for sites with high herbivore intensity, where the effects of herbivores on biomass were strong. Differences in herbivore diet and density across our study sites produce a gradient in the potential site-level control of forbs by herbivores[26,28,32]. Herbivore diet, for example, can determine how herbivory modulates grass-forb competition[26,28]. Herbivores that predominantly consume grasses may benefit forbs via relaxing light competition between grasses and forbs; the loss of these large grazers can negatively affect forb abundance and diversity[8,26,31,32]. Alternatively, as some forbs are palatable and are preferred forage for many herbivores[55,56], the loss of forb-consuming herbivores can instead increase forb diversity and abundance[25,32]. Our results provide evidence for these impacts: at sites with greatest herbivore effects on total biomass, we find that herbivores increase both richness and cover of abundant forb families while suppressing grass cover. Variation in herbivore intensity may modulate the outcomes of grass-forb competition and coexistence[25,57,58].

In contrast to our third prediction and some prior findings[38,59], PET did not interact with fertilization and herbivore exclusion. Instead, increasing PET decreased forb richness, but did not affect grass cover and richness. This finding suggests that the increase in grass cover under fertilization may be consistent across climatic variation in aridity, while aridity may further drive forb declines. This result is supported by prior work that found that forb functional diversity decreased under increasing aridity, and that variation in rainfall modulated grass-forb coexistence[14,60]. In contrast, other studies have found forbs to be more resilient under dry conditions than grasses and that increased precipitation increased grass biomass[46], perhaps due to the differing ways in which these studies have measured aridity[37,59,61]. Our results from 94 grasslands across six continents suggest, however, that aridity generally has negative effects on forb diversity.

In support of our fourth prediction, fertilization effects further varied by forb taxonomic family. Fabaceae and Asteraceae declined under nitrogen enrichment, while Geraniaceae declined under phosphorus enrichment, Polygonaceae declined under enrichment of potassium with micronutrients, and Apiaceae did not respond to nutrient effects, consistent with past findings[10,62]. While Fabaceae increased under enrichment of phosphorus and potassium with micronutrients, Polygonaceae increased under nitrogen enrichment. Forb families differ in floral traits and floral rewards, suggesting that compositional shifts in forb families under fertilization could have impacts upon communities of pollinators that depend on these forbs[63].

Declines in forbs under nutrient enrichment may have consequences for ecosystem functions and services, as forbs constitute a large portion of functional diversity in grasslands[5,60]. Forbs play a critical role in contributing to grassland species richness, ecosystem functions, and ecological stability,

**Table 2 | Fertilization factorial results**

| Predictors | Forb Species Richness | | Forb Family Richness | | Forb Cover | | Grass Species Richness | | Grass Cover | |
|---|---|---|---|---|---|---|---|---|---|---|
| | Estimates | t-value; p-value | Estimates | t-value; p-value | Estimates | t-value; p-value | Estimates | t-value; p-value | Estimates | t-value; p-value |
| Intercept | 0.01 (−0.06 to 0.08) | 0.38; 0.704 | −0.01 (−0.07 to 0.05) | −0.19; 850 | 0.04 (−0.11 to 0.19) | 0.53; 0.594 | −0.01 (−0.07 to 0.05) | −0.40; 0.689 | −0.03 (−0.14 to 0.07) | −0.64; 0.525 |
| N | −0.15 *** (−0.20 to −0.10) | **−6.12; <0.001** | −0.08 *** (−0.13 to −0.04) | **−3.49; <0.001** | −0.07 ** (−0.12 to −0.03) | **−3.18; 0.001** | −0.01 (−0.06 to 0.03) | −0.67; 0.503 | 0.08 *** (0.05 to 0.12) | **4.44; <0.001** |
| P | −0.06 * (−0.11 to −0.01) | **−2.32; 0.020** | −0.01 (−0.06 to 0.03) | −0.54; 0.592 | −0.01 (−0.05 to 0.04) | −0.36; 0.715 | −0.01 (−0.05 to 0.03) | −0.59; 0.557 | 0.10 *** (0.06 to 0.13) | **5.29; <0.001** |
| Kμ | −0.06 ** (−0.11 to −0.02) | **−2.59; 0.010** | −0.04 (−0.09 to 0.00) | −1.78; 0.075 | −0.00 (−0.05 to 0.04) | −0.10; 0.921 | 0.00 (−0.04 to 0.04) | 0.11; 0.911 | −0.01 (−0.05 to 0.02) | −0.78; 0.438 |
| N × P | −0.00 (−0.07 to 0.07) | −0.06; 0.951 | −0.03 (−0.09 to 0.04) | −0.80; 0.421 | | | 0.02 (−0.04 to 0.08) | 0.60; 0.552 | | |
| N × Kμ | 0.05 (−0.02 to 0.12) | 1.39; 0.165 | 0.05 (−0.02 to 0.11) | 1.44; 0.149 | | | −0.00 (−0.06 to 0.06) | −0.01; 0.991 | | |
| P × Kμ | 0.02 (−0.05 to 0.09) | 0.61; 0.540 | −0.03 (−0.09 to 0.04) | −0.83; 0.405 | | | 0.04 (−0.02 to 0.10) | 1.27; 0.204 | | |
| N × P × Kμ | −0.05 (−0.15 to 0.04) | −1.07; 0.285 | −0.03 (−0.12 to 0.06) | −0.67; 0.500 | | | −0.14 ** (−0.22 to −0.05) | **−3.10; 0.002** | | |
| **Random Effects** | | | | | | | | | | |
| σ2 | 0.26 | | 0.22 | | 0.86 | | 0.17 | | 0.49 | |
| τ00 | 0.28 block | | 0.23 block | | 0.63 block | | 0.24 block | | 0.45 block | |
| | 0.12 site_code | | 0.11 site_code | | 0.25 site_code | | 0.09 site_code | | 0.23 site_code | |
| N | 6 block | | 6 block | | 6 block | | 6 block | | 6 block | |
| | 84 site_code | | 84 site_code | | 84 site_code | | 87 site_code | | 87 site_code | |
| Observations | 13,686 | | 13,684 | | 13,686 | | 14,259 | | 14,259 | |

Mixed effects model results for the effects of fertilization by nutrient type on Forb Family Richness, Forb Species Richness, Normalized Forb Cover, Grass Species Richness, and Normalized Grass Cover. The intercept is the mean value of the unfenced and unfertilized control plots.

All response variables were calculated using LRRs. The parenthetical numbers are the confidence interval. Bolded numbers indicate statistical significance. $N$ = 89 biologically independent sites.

$* p < 0.05 ** p < 0.01 *** p < 0.001$

**Fig. 3 | Nutrient effects by family.** The effect of different nutrients (fertilization factorial experiment) on (**A**) species richness and (**B**) family-level cover for Asteraceae (yellow), Fabaceae (indigo), Geraniaceae (pink), Apiaceae (magenta) and Polygonaceae (brown). Model estimates are shown relative to the control treatment (estimate = 0). Response variables were converted to log response variables to account for the change from pretreatment to current data with cover data normalized by maximum plot cover. Multiple nutrient interactions are included for richness. *N* refers to nitrogen, *P* refers to phosphorus, and Kμ refers to Potassium with micronutrients. Error bars show the 89% confidence interval. Asteraceae (*n* = 89 sites), Fabaceae (*n* = 85 sites), Geraniaceae (*n* = 28 sites), Apiaceae (*n* = 46 sites), and Polygonaceae (*n* = 54 sites).

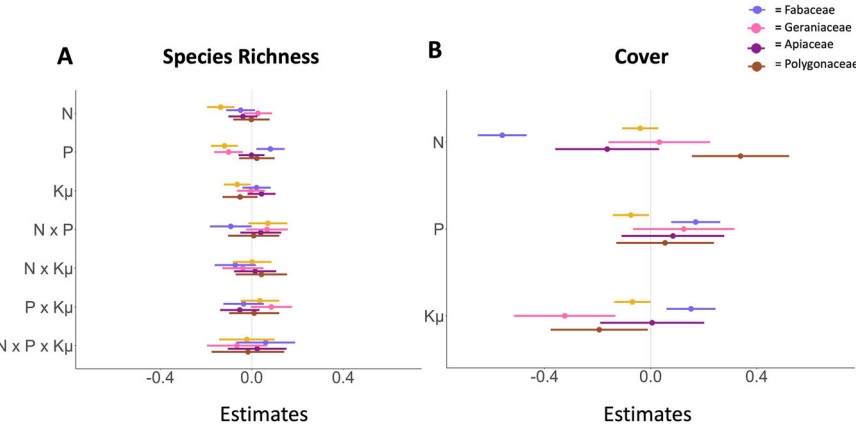

such as nutrient cycling and provision of food for pollinators[60]. A greater diversity of forbs may provide both more functional redundancy and niche complementarity that could make forb communities more resilient to disturbance[64,65] and benefit pollinators[66]. A loss in forb richness due to nutrient enrichment may thus reduce the resiliency of grassland forb communities to further perturbations under anthropogenic global change, for example, due to climate change. Restoration efforts could consider maximizing the phylogenetic and functional diversity of forbs while mitigating nutrient enrichment and reintroducing or protecting the local, native large mammalian herbivores[3,67–69].

Forb declines under anthropogenic change can have downstream effects on plant-pollinator mutualisms[39,41,70,71]. The shift from forb to grass-dominated, less forb-rich ecosystems likely has negative functional consequences for pollinators that require a diversity and abundance of forbs as floral resources[4,72,73]. Our findings of strong nutrient enrichment effects on Fabaceae and Asteraceae are especially concerning, giving that these two abundant families respectively provide critical early and late season resources for pollinators[47]. Fertilization can alter the quality[74,75] and quantity[41,76,77] of floral resources that forbs provide to pollinators and can shift floral phenology[78,79]. As decreases in forb richness may decrease pollination services essential for food production and agriculture[41,80], future research could examine how nutrient-driven forb declines affect pollinators and whether grazing can rescue these forb-pollinator mutualisms. These widespread fertilization effects on forbs, modulated by herbivory may be pivotal in explaining pollinator declines at a global scale[80–82]. This research supports past findings that large herbivores can rescue forbs from the negative effects of nutrient enrichment and that nitrogen enrichment, in particular, is detrimental to forbs[28,46,58]. These findings further suggest that these responses are generalizable to the global scale and that forb response varies by taxonomic family with nitrogen enrichment strongly decreasing Fabaceae and Asteraceae but not Apiaceae or Gerianaceae, and increasing Polygonaceae.

## Methods
### Study design
This study used data from 94 sites in the Nutrient Network (https://nutnet.org/), an experimental study of nutrient enrichment and herbivore exclusion in grasslands that is globally replicated[8]. At each site, 5 × 5 m plots included a factorial combination of nutrient additions of nitrogen (N), phosphorus (P), potassium (K), and micronutrients (Kμ) or no nutrients (control) and fencing or not to yield 10 treatment plots per block (for more details on experimental design see Borer et al. 2014). Most sites contained three replicate blocks. Site latitude, longitude, mean Potential Evapotranspiration (PET), year of treatment applications, and related metadata are detailed in Supplementary Data 1. The following nutrients were added annually to the fertilized plots: 10 g $Nm^{-2}y^{-1}$ as slow-release urea ($(NH_2)_2CO$), 10 g $Pm^{-2}y^{-1}$ as triple-super phosphate ($Ca(H_2PO_4)_2$), 10 g $Km^{-2}y^{-1}$ as

potassium sulfate ($K_2SO_4$). The plots receiving the potassium treatment received a one-time addition of other micronutrients and macronutrients in the first year: 100 gm-2 of a mixture of 15% iron (Fe), 14% sulfur (S), 1.5% magnesium (Mg), 2.5% manganese (Mn), 1% copper (Cu), 1% zinc (Zn), 0.2% boron (B) and 0.05% molybdenum (Mo). The control plots were left untreated. Herbivore exclusion fences were up to 2.3 m high with the goal of excluding all aboveground large mammalian herbivores more than 50 g, including rabbits, hares, and marsupials[8,28,32]. Sites varied in their climate[8], soil fertility[44], species richness and composition, and grazing history[28].

We analyzed data from these 10 experimental plots in two combinations. First, "fertilization factorial" plots (*n* = 89 sites) applied different factorial combinations of nitrogen, phosphorus, and potassium to experimental plots: control, N, P, Kμ, NP, PKμ, NKμ, and NPKμ with the control plots left unfertilized[8]. Second, "fencing by fertilization" plots (*n* = 82 sites) combined large herbivore exclusion using fencing with NPKμ fertilization resulting in four treatments: unfenced control, fenced control, unfenced NPKμ, and fenced NPKμ[8].

### Vegetation sampling
Sampling at all sites followed a standardized sampling protocol[8]: all plots were 5 × 5 m, and all sites collected at least one year of pre-treatment data and at least two years of post-treatment data[8]. The plots were non-destructively sampled for vegetative cover; a 1 × 1 m quadrat was used to estimate aerial vegetative cover within each plot for each plant species. Summed cover may exceed 100% if vegetation contains multiple layers. Site scientists provided information on functional lifeforms (ex: graminoid, forb, woody, etc). For sites where cover was assessed multiple times each year, species were assigned their maximum cover across the different dates. Annual peak season live biomass was measured as the aboveground live biomass of all plants rooted within two 10 × 100 cm strips per plot[8,32]. Clipped vegetation was dried to a constant mass at 60 °C for 48 h, and then weighed to the nearest 0.01 g[8,32].

### Statistics and reproducibility
All data analyses were performed in R version 4.4.1[83] using the "nlme"[84], "lme4"[85], "sjPlot"[86], and "Rmisc"[87] packages. We examined responses for combined legume and forb functional groups (hereafter referred to as forbs) and combined grasses and graminoids (hereafter referred to as grasses) for 94 grassland sites: 89 sites had a fertilization factorial experiment, while 82 sites had a fencing by fertilization experiment. We first calculated the following diversity metrics: forb and grass species richness (total number of species present), and forb family richness (total number of families present) for each plot and year per site. We calculated the total normalized forb vegetative cover per plot for each year (total forb cover/total plot vegetative cover) and normalized grass cover for each plot in each year (total grass cover/total plot vegetation cover) for a given year at each site. For the five most abundant

taxonomic families of forbs across our dataset, Asteraceae ($n = 89$ sites), Fabaceae ($n = 85$ sites), Geraniaceae ($n = 28$ sites), Apiaceae ($n = 46$ sites), and Polygonaceae ($n = 54$ sites) (Tables S2, S3), we calculated response variables of species richness within a family and total normalized forb cover within a family (cover of family in plot/total plot vegetative cover) for a given year at each site.

For all response variables, we calculated a log response ratio (LRR) that accounted for differences in pre- (i.e., year 0) and post-treatment data. We used the formula ln(Experimental Treatment data/Pretreatment data) for each plot sampled each year. We used pretreatment data in the denominator to account for initial site variation prior to treatments. We then calculated the percent change using the formula $100 \times (e^{LRR} - 1)$ where the LRR was the model estimate since the natural log was used to calculate LRR. An alpha level of $p = 0.05$ was the threshold for statistical significance.

To test our predictions, we used separate linear mixed effects models for all response variables with global change treatments as fixed effects (described below). Our models included a random intercept of blocks nested within site. To account for autocorrelation of responses within plots since treatments began, we included a corAR1 autocorrelation-moving average temporal correlation structure of years since treatments began nested by site, block, and plot. We further calculated model significance using a type III ANOVA to account for random effects and autocorrelation structures using the "car" package in R (see Supplemental Tables S11-S14)[88].

To test our first prediction about type of fertilization, we ran these linear mixed effects models for all response variables with N, P, and Kμ as fixed effects. Interactions between climate and fertilization were non-significant, so we did not include climate in these models. We took advantage of our full factorial experimental design to explore all interactions of our experimental treatment fixed effects (e.g., N*P*Kμ) for our models with richness and cover as response variables. In these models, these nutrient predictor variables tended to show significant two- or three-way interactions for models of richness, but not for models of cover with the exception of of Fabaceae and Polygonaceae (see Table S11, Supplementary Data 2). However, there is still information to be gained from the data despite non-significant interactions for our predictor variables. Thus, for our models where cover was our response variable, we dropped all interactions and focused on additive models of our predictor variables (N, P, and Kμ). This backward selection approach allowed us to leverage our experimental design to more fully quantify the controls on forb and grass richness and abundance.

To test our second and third predictions about herbivory and aridity, we ran linear mixed effects models for all response variables with fixed effects of fencing and NPKμ treatments, herbivore intensity, and potential evapotranspiration (PET). We included interactions between NPKμ, fencing, and herbivore intensity as well as interactions between NPKμ, fencing, and PET. We calculated site-level herbivore intensity as the mean difference in live biomass between unfenced and fenced controls (Unfenced Control Live Biomass to Fenced Control Live Biomass) in year 1 of treatments for each block and then took the mean herbivore intensity across blocks at the site level[32], such that increasingly negative values of this metric indicate greater reduction of herbivore intensity under herbivore exclusion, reflecting the greater impact of herbivore-exclusions on site-level biomass. We extracted average yearly PET data from the CRU climate dataset for each year through 2016[89], a measure of aridity. We selected PET because it combines Mean Annual Precipitation (MAP) and Mean Annual Temperature (MAT). We scaled PET for each year.

## Reporting summary
Further information on research design is available in the Nature Portfolio Reporting Summary linked to this article.

## Data availability
Source data and metadata associated with this paper are archived in the following publically accessible Environmental Data Initiative (EDI) repository ID edi.1823.2: https://doi.org/10.6073/pasta/62e2c0f1bc1ccb5a 29d63b513bb66810. Deposited data can be accessed via the link provided under the repository 'Data for Forb diversity globally is harmed by nutrient enrichment but can be rescued by large mammalian herbivory'[90]. Please contact the corresponding author for further information. Data can be cited as follows: Barrio, I., E. Boughton, C. Chu, G. Du, Q. Li, W. Li, G. Wen, N. Eisenhauer, S. Haider, J. Siebert, K. Speziale, D. Wedin, A. Jentsch, M. Spohn, K. Davies, B. Melbourne, B. Mortensen, J. Paper, E. Borer, L. Hallett, J. Firn, Y. Buckley, I. Donohue, L.A. Biederman, K.S. Hofmockel, L. Sullivan, A. Kay, J.M. Knops, E. Chaneton, P.M. Tognetti, L. Yahdjian, M. Bugalho, M. Caldeira, A. MacDougall, K. Elgersma, R. Laungani, E. Cleland, G. Wardle, S. Güsewell, Y. Hautier, A. Hector, K.P. Kirkman, M. Tedder, J. Nelson, N.M. DeCrappeo, D. Pyke, M.J. Crawley, K.L. Cottingham, E.M. Wolkovich, J. Zinnert, C.S. Brown, K. Jamiyansharav, A. Lkhagva, A. Ebeling, C. Roscher, L. Brudvig, M. Sankaran, A. Richards, A. Eskelinen, R. Virtanen, J. Morgan, M. Cadotte, A. Weiss, L. Lannes, H. olde Venterink, C. Stevens, L. Hallett, N. Smith, J. Alberti, P. Daleo, H. Martinson, B. Osborne, S. Reed, M. DuPre, K. Laflamme, Y. Lekberg, A. Wallace, S.M. Prober, M. Akasaka, T. Kadoya, J. Catford, H. Hillebrand, S. Baez, J. Price, R. Standish, J. Dwyer, H. Bahamonde, P. Peri, A. Eskelinen, D.S. Gruner, L. Yang, K.J. Komatsu, M. Smith, S. Koerner, A. Young, L. Brudvig, C.M. D'Antonio, E. Seabloom, T.M. Anderson, S. Collins, L. Ladwig, D.M. Blumenthal, C.S. Brown, J.A. Klein, A. Knapp, P. Adler, W.S. Harpole, J.D. Bakker, J. Hille Ris Lambers, R.L. McCulley, P.D. Wragg, D. Orr, H. Young, P.A. Fay, J. Martina, A. Leakey, E.I. Damschen, T. Knight, J.L. Orrock, K.P. Kirkman, M. Tedder, C. Mitchell, J. Wright, N. Pichon, A.C. Risch, M. Schuetz, R. Mitchell, R. Ochoa Hueso, S. Power, and R. Nelson. 2025. Data for Forb diversity globally is harmed by nutrient enrichment but can be rescued by large mammalian herbivory ver 2. Environmental Data Initiative. https://doi.org/10.6073/pasta/62e2c0f1bc1ccb5a29d63b513bb66810.

## Code availability
Code associated with this paper is archived in the following publically accessible Zenodo repository 14207290[91].

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

## Acknowledgements

Author contributions, site acknowledgments, and funding acknowledgments are documented in Supplementary Data 3. The authors would like to thank Dr. Neal Williams and Dr. Fernanda Valdovinos for providing additional feedback on this manuscript as well as the editors and peer reviewers. The authors would like to thank the Dirzo Lab at Stanford University for providing input on this research. This work was generated using data from the Nutrient Network (http://www.nutnet.org) experiment, funded at the site-scale by individual researchers (see Supplemental Data 3). Coordination and data management have been supported by funding to E. Borer and E. Seabloom from the National Science Foundation Research Coordination Network (NSF-DEB-1042132) and Long Term Ecological Research (NSF-DEB-1234162 and NSF-DEB-1831944 to Cedar Creek LTER) programs, and the Institute on the Environment (DG-0001-13). We also thank the Minnesota Supercomputer Institute for hosting project data and the Institute on the Environment for hosting Nutrient Network meetings.

## Author contributions

Author contributions, site acknowledgments, and funding acknowledgments are documented in detail in Supplementary Data 3. Author contributions are listed in accordance with the Nutrient Network guidelines for authorship. Developed and framed research question(s): R.A.N., E.H.G., S.P. Harrison, E.B., E.S., and L.S. Gave input on paper storyboard/concepts: R.A.N., A.D., H.O.V., A.S.M., E.H.G., M.G., E.B., E.S., L.C., S.P., N.J., G.W., R.V., C.J.S., W.S.H., P.M.T., A.Y., J.P.C., N.G.S., P.G., X.R., N.H., A.J., M.S., D.O., P.L.P., J.A.C., K.J.E., A.C.R., L.B., J.W.M., A.A.W., I.D., S.E.K., D.S.G., B.B.O., Y.H., H.M., L.S.L., and L.S. Analyzed data: RAN. Contributed to data analyses:

R.A.N., A.D., E.H.G., P.M.T., and L.S. Wrote the paper: R.A.N. Contributed to paper writing: R.A.N., A.D., E.H.G., E.B., L.C., S.P., G.W., P.B.A., A. Eskelinen, R.V., N.E., C.E., C.J.S., W.S.H., P.M.T., A.Y., N.G.S., P.G., X.R., N.H., A.J., M.S., D.O., P.L.P., J.A.C., K.J.E., A.C.R., L.B., M.C.C., J.W.M., A.A.W., I.D., S.E.K., D.S.G., B.B.O., Y.H., H.M., L.L., M.N.B., A. Ebeling, G.F.V., L.S., C.R.D., S.P. Haider, and S.P. Harrison. Nutrient Network Site Coordinator: H.O.V., A.S.M., E.B., E.S., S.P., G.W., P.B.A., A. Ekselinen, R.V., N.E., C.E., C.J.S., P.M.T., A.Y., N.G.S., X.R., N.H., A.J., D.O., P.L.P., J.A.C., K.J.E., A.C.R., L.B., M.C.C., J.W.M., A.A.W., I.D., S.E.K., D.S.G., B.B.O., Y.H., H.M., L.S.L., M.N.B., A. Ebeling, G.F.V., L.S., C.R.D., and S. Haider. Nutrient Network Coordinator: E.B., E.S., and W.S.H.

## Competing interests

The authors declare no competing interests.

## Ethical approval

The authors collected data after receiving appropriate permissions to access sites. To ensure author inclusion, author and data contributor contributions followed the guidelines of the Nutrient Network. See https://nutnet.org/ for more details.

## Additional information

[1]University of California, Davis, Department of Environmental Science & Policy, Davis, CA, USA. [2]Department of Wildland Resources and the Ecology Center, Utah State University, Logan, UT, USA. [3]Department of Plant Biology, Michigan State University, East Lansing, MI, USA. [4]W K Kellogg Biological Station, Michigan State University, Hickory Corners, East Lansing, MI, USA. [5]Ecology, Evolution and Behavior Program, Michigan State University, East Lansing, MI, USA. [6]Michigan Technological University, Dept. of Biological Sciences, Houghton, MI, USA. [7]Dept. of Ecology, Evolution, and Behavior, University of Minnesota, St. Paul, MN, USA. [8]Instituto de Investigaciones Fisiológicas y Ecológicas Vinculadas a la Agricultura (IFEVA), Facultad de Agronomía, Universidad de Buenos Aires y CONICET, Buenos Aires, Argentina. [9]Iowa State University, Ames, IA, USA. [10]Center for Applied Ecology "Prof. Baeta Neves" (CEABN-InBIO), School of Agriculture, University of Lisbon, Lisbon, Portugal. [11]Forest Research Centre, Associate Laboratory TERRA, School of Agriculture, University of Lisbon, Lisbon, Portugal. [12]Centre for Ecology, Evolution and Environmental Changes (cE3c)/Azorean Biodiversity Group & University of Azores, Departamento de Ciências e Engenharia do Ambiente, Angra do Heroísmo, Azores, Portugal. [13]Departamento de Ecologia, Universidade Federal de Goiás, Goiânia, Brazil. [14]Department of Geography, King's College London, London, United Kingdom. [15]Fenner School of Environment & Society, Australian National University, Canberra, ACT, Australia. [16]School of Life and Environmental Sciences, University of Sydney, Sydney, NSW, Australia. [17]Department of Integrative Biology, University of Guelph, Guelph, ON, Canada. [18]Zoology, School of Natural Sciences, Trinity College Dublin, Dublin, Ireland. [19]Institute for Biodiversity, Ecology and Evolution, University Jena, Jena, Germany. [20]German Centre for Integrative Biodiversity Research (iDiv), Leipzig, Germany. [21]Leipzig University, Leipzig, Germany. [22]Department of Biology, University of Northern Iowa, Cedar Falls, IA, USA. [23]Ecology and Genetics Unit, University of Oulu, Oulu, Finland. [24]Department of Life Sciences, Imperial College London, Silwood Park, London, United Kingdom. [25]Department of Animal and Range Sciences, New Mexico State University, Las Cruces, NM, USA. [26]Agencia de Extensión Rural Coronel Suárez, EEA Cesáreo Naredo, Instituto Nacional de Tecnología Agropecuaria (INTA), Coronel Suárez, Buenos Aires, Argentina. [27]Department of Entomology, University of Maryland, College Park, MD, USA. [28]Mammal Research Institute, Department of Zoology & Entomology, University of Pretoria, Pretoria, South Africa. [29]Institute of Ecology, Faculty of Sustainability, Leuphana University of Lüneburg, Lüneburg, Germany. [30]Helmholtz Center for Environmental Research Ð UFZ, Department of Physiological Diversity, Leipzig, Germany. [31]Martin Luther University Halle-Wittenberg, Halle (Saale), Germany. [32]Ecology and Biodiversity Group, Department of Biology, Utrecht University, Utrecht, CH, The Netherlands. [33]Disturbance Ecology and Vegetation Dynamics, Bayreuth Center of Ecology and Environmental Research, University of Bayreuth, Bayreuth, Germany. [34]Department of Biology, University of North Carolina Greensboro, Greensboro, NC, USA. [35]Department of Biology and Animal Sciences, São Paulo State University - UNESP, Ilha Solteira, SP, Brazil. [36]Department of Biology, McDaniel College, Westminster, MD, USA. [37]Department of Environment & Genetics, La Trobe University, Bundoora, Melbourne, VIC, Australia. [38]Department Biology, Vrije Universiteit Brussel (VUB), WILD, Brussels, Belgium. [39]USDA ARS, Eastern Oregon Ag Research Center, Burns, OR, USA. [40]Department of Environmental and Society, Utah State University, Moab, UT, USA. [41]Instituto Nacional de Tecnologia Agropecuaria (INTA), Universidad Nacional de la Patagonia Austral (UNPA), Rio Gallegos, Argentina. [42]Hawkesbury Institute for the Environment, Western Sydney University, Sydney, NSW, Australia. [43]Sorbonne Université, CNRS, IRD, INRA, Université Paris Cité, UPEC, Institute of Ecology and Environmental Sciences—Paris, Paris, France. [44]Swiss Federal Institute for Forest, Snow and Landscape Research WSL, Birmensdorf, Switzerland. [45]Department of Life Science, National Taiwan University, Taipei, Taiwan. [46]Department of Biological Sciences, Texas Tech University, Lubbock, TX, USA. [47]Lancaster Environment Centre, Lancaster University, Lancaster LA1 4YQ, United Kingdom. [48]Netherlands Institute of Ecology, Wageningen, The Netherlands. [49]Department of Integrative Biology, University of Texas at Austin, Austin, TX, USA. ✉e-mail: becca.nelson@usu.edu

