## [Transparent Peer Review file · Communications Biology]

Forb diversity globally is harmed by nutrient enrichment but can be rescued by large mammalian herbivory

Corresponding Author: Dr Rebecca Nelson

Version 0:

Reviewer comments:

Reviewer #1

(Remarks to the Author)

This study investigates the global patterns of forb dynamics in grassland ecosystems through a factorial experiment conducted at 94 sites across six continents. The research aims to test the generality of three key predictions derived from previous studies. Results showed that nutrient addition reduces forb cover and richness, with nitrogen having the greatest impact. Large mammal herbivory enhances forb cover only when nutrients are enriched and herbivore intensity is high. Additionally, forb richness is lower in more arid sites but not influenced by climate-nutrient or climate-herbivory interactions. The research also highlights that nitrogen enrichment disproportionately affects certain forb families (e.g., Asteraceae, Fabaceae).

The manuscript is broadly well-written, and nicely introduces key ideas on forb dynamics in grassland ecosystems responses to environmental change. It addresses good, clear subjects to test the generality of three key predictions derived from previous studies. These results have important implications for understanding the response of grassland ecosystem to environmental changes. Overall, the manuscript can make a nice contribution to the literature, but I believe that the concerns below first need to be addressed.

My main overarching concern pertains to the method of statistical analyses: I noticed that the authors calculated the response ratio using the differences in pre- and post-treatment data, rather than the more commonly used differences in treatment and control treatment data. Why is that? How can the impact of varying climate conditions across different years and sites be avoided? I did not find any explanation or clarification regarding this in the methods section. I highly recommend using pretreatment data as random factors to account for initial site variation, rather calculating the "RR".

Does the dominance of forbs vary across the sites involved in this study? Please provide relevant data. It is evident that the dominance of forbs is a key factor in determining their response to environmental changes. Therefore, I believe that the influence of forb dominance at different sites should be considered during the data analysis, potentially as a random factor.

In addition, the generalizability of the findings and the comparison with other studies can be expanded. You can also mention what kind of things might be interesting to do in the future (longer time series, etc.). This study claims to have integrated data from 94 sites across 6 continents in an attempt to test the global generality of previous findings. However, the discussion and analysis in this paper did not mention the multiple global synthesizes published on the effects of nutrient enrichment or grazing on grassland biodiversity in recent years. Hence, you have to complete the discussion with the results that are now lacking, giving more detailed explanations and interpretations and comparing your results with previous synthesizes results, and probably mentioned them in introduction section.

L358 treatments repeated;
L377 with repeated ;

Li W, Gan X, Jiang Y, Cao F, Lü X-T, Ceulemans T, Zhao C (2022) Nitrogen effects on grassland biomass production and biodiversity are stronger than those of phosphorus. *Environmental Pollution*, 309, 119720.

Reviewer #2

(Remarks to the Author)

A very solid study from the global NutNet Experiment that gives valuable insights into the interactive effects of multiple global change drivers on vegetation i.e. eutrophication, land use change (e.g. grazing) and drought.

I have only minor comments regarding the clarity of presentation which I attach as in-line comments in the main manuscript file.

Best regards.

Version 1:

Reviewer comments:

Reviewer #1

(Remarks to the Author)

This resubmitted manuscript has been well improved after revision.

Reviewer #2

(Remarks to the Author)

Dear authors,

I enjoyed reading the revised version of your manuscript and am overall satisfied with your amendments. A valuable publication with insightful results, which I will share with my students for sure, congratulations!

I was left with 2 tiny comments and one statistical question:

Comments:

L353: herbivory offsets the negative effects of fertilization

L360: ... as some forbs are ... (space was missing between some and forbs)

Question:

The model description is very transparent and I agree with all measures you undertook for handling spatial, sampling-design-related and temporal dependencies in the data. The only aspect I was missing was a note on which Anova type you used for testing the significances of effects maybe that is worth adding. Given that you used interaction terms, you probably used type I or better type III Anova. I would use type III as this will account for random effects and the temporal AR structure when testing the effect of any fixed effect (`car::Anova(..., type="III")`). And just to confirm, as I have been dealing with this lately, you retrieved the p-values from the Anova object in R as this uses the appropriate/chooses test for your data set (random effects etc). The summary object gives the estimates, but uses only t-tests to estimate the sign. Difference of estimates compared to 0 and with interactive categorical factors only shows comparisons to one factor level. See recommendation by Zuur et al. "Introduction to Mixed-Effects Models".

Best wishes

Lena Neuenkamp

Version 2:

Reviewer comments:

Reviewer #2

(Remarks to the Author)

Dear authors,

thank you for clarification on the significance testing, with which I fully agree.

I have no further comments. Congratulations to this very insightful study!

Best wishes

Lena Neuenkamp

We therefore invite you to revise and resubmit your manuscript, taking into account the points raised. In particular, please note that the following revisions would be necessary for us to contact our referees again:

1 - Please address the concerns raised by reviewer #1 about the statistical analysis.

2 - As suggested by reviewer #1, please include additional data about the influence of forb dominance at different sites.

Additional comment from the editor I: Please note that the comments from reviewer #2 can be found in the file attached to this email.

Additional comment from the editor II: We note that the reviewers have indicated to cite specific publications. As always, we would ask you to carefully assess whether these publications are relevant to your study and decide whether to discuss and cite these references based on your own assessment. Please note that citing these publications is not mandatory.

Please highlight all changes in the manuscript text file.

Reviewers' comments:

Dear Reviewers:

Thank you for the helpful suggestions. Please find attached an updated version of our manuscript that incorporates and addresses all of these suggestions. We have highlighted our edits to the body of the manuscript in tracked changes.

Reviewer #1 (Remarks to the Author):

This study investigates the global patterns of forb dynamics in grassland ecosystems through a factorial experiment conducted at 94 sites across six continents. The research aims to test the generality of three key predictions derived from previous studies. Results showed that nutrient addition reduces forb cover and richness, with nitrogen having the greatest impact. Large mammal herbivory enhances forb cover only when nutrients are enriched and herbivore intensity is high. Additionally, forb richness is lower in more arid sites but not influenced by climate-nutrient or climate-herbivory interactions. The research also highlights that nitrogen enrichment disproportionately affects certain forb families (e.g., Asteraceae, Fabaceae).

The manuscript is broadly well-written, and nicely introduces key ideas on forb dynamics in grassland ecosystems responses to environmental change. It addresses good, clear subjects to test the generality of three key predictions derived from previous studies. These results have important implications for understanding the response of grassland ecosystem to environmental changes. Overall, the manuscript can make a nice contribution to the literature, but I believe that the concerns below first need to be addressed.

My main overarching concern pertains to the method of statistical analyses: I noticed that the

authors calculated the response ratio using the differences in pre- and post-treatment data, rather than the more commonly used differences in treatment and control treatment data. Why is that? How can the impact of varying climate conditions across different years and sites be avoided? I did not find any explanation or clarification regarding this in the methods section. I highly recommend using pretreatment data as random factors to account for initial site variation, rather than calculating the “RR”.

Thank for these suggestions. We used a log-response ratio of the difference in pre vs post treatment data to account for variation in initial site conditions such as forb dominance. We ran a similar set of analyses in which we used the difference in LRR between the treatment and control using the methods from Lind et al. 2013, and found that our results were the same, so we chose to use the pre vs post-treatment LRR for simplicity. To incorporate your suggestion, we have now included these results for the difference in LRR between the treatment and the control in Supplemental Tables 8 and 9.

We agree it is important to account for site-level and year to year variation in environmental conditions. Because climate is often autocorrelated across years, we use an autocorrelation-moving average temporal correlation structure (AR1), which accounts for how much temporal data are correlated with their own past values at a site. Thus, a moving average temporal correlation structure can account for the impact of varying climate conditions across different years and sites. We describe this approach in Lines 390-393: “To account for autocorrelation of responses within plots since treatments began, we included a corAR1 autocorrelation-moving average temporal correlation structure of years since treatments began nested by site, block, and plot” and have updated our language to clarify this.

To further account for variation among sites, we use a random effect term that accounts for variation among sites. This approach also accounts for differences in initial forb dominance across sites. We explain this approach in Lines 389-390: “Our models included a random intercept of block nested within site.”

Our pretreatment data are numerical, continuous variables. The random effects structure we use can only use categorical variables for random effects (Bates et al. 2015). Therefore, we chose to use an LRR approach instead along with a random effect of site and an autocorrelation-moving average temporal correlation structure.

**Bates, D., Mächler, M., Bolker, B., & Walker, S. (2015). Fitting Linear Mixed-Effects Models Using lme4. *Journal of Statistical Software*, 67(1), 1–48.
<https://doi.org/10.18637/jss.v067.i01>**

**Lind, Eric M., et al. (2013) Life-history constraints in grassland plant species: a growth-defence trade-off is the norm." *Ecology letters* 16(4), 513-521.
<https://doi.org/10.1111/ele.12078>**

Does the dominance of forbs vary across the sites involved in this study? Please provide relevant data. It is evident that the dominance of forbs is a key factor in determining their response to environmental changes. Therefore, I believe that the influence of forb dominance at different sites should be considered during the data analysis, potentially as a random factor.

We agree that it is important to consider the effects of forb dominance, as it does vary across sites. To address this, we have provided data on pretreatment forb dominance at each site in Supplemental Table 10. Moreover, Supplemental Table 3 lists out the type of grassland present at each site (e.g. tallgrass prairie, heathland, old field etc), which may provide further insight into the ecological role of forbs at each site.

To account for forb dominance as well as initial site conditions, we used the formula $\ln(\text{Experimental Treatment data}/\text{Pretreatment data})$ for each plot sampled each year. For each response variable, we used pretreatment data in the denominator to account for initial forb dominance. For example, for a hypothetical plot that initially has a value of 20 for normalized forb cover in the pretreatment data, if the post-treatment data had a cover value of 7, the log-response ratio would be $\ln(7/20)$. In contrast, if another hypothetical plot that has initial value of 5 for forb cover and the post-treatment plot has similarly has a cover value of 7, we would get $\ln(7/5)$. Thus, the log response ratio scales the magnitude of the response variables in relation to their initial, pre-treatment plot conditions.

Because normalized forb cover, our measure of forb dominance, is a continuous variable, we cannot use it as a random effect (see Bates et al. 2015 for a statistical explanation for why this is). Instead we choose to use site, a categorical variable that is thus compatible with a random effects structure, as a random effect, to account for site-level differences in forb dominance to address your concern.

Furthermore, our colleagues Wilfahrt et al. 2023 have already published on how dominant species respond to these global changes using our same Nutrient Network dataset, finding that initially dominant forb species are still sensitive to the effects of herbivore exclusion and nutrient enrichment. We now reference these findings from Wilfahrt et al. 2023 in the Introduction and Discussion of our paper in lines 144 and 249.

**Bates, D., Mächler, M., Bolker, B., & Walker, S. (2015). Fitting Linear Mixed-Effects Models Using lme4. *Journal of Statistical Software*, 67(1), 1–48.
<https://doi.org/10.18637/jss.v067.i01>**

Wilfahrt, Peter A., et al. (2023) "Nothing lasts forever: Dominant species decline under rapid environmental change in global grasslands." *Journal of Ecology* 111(11), 2472-2482.

In addition, the generalizability of the findings and the comparison with other studies can be expanded. You can also mention what kind of things might be interesting to do in the future (longer time series, etc.). This study claims to have integrated data from 94 sites across 6 continents in an attempt to test the global generality of previous findings. However, the

discussion and analysis in this paper did not mention the multiple global syntheses published on the effects of nutrient enrichment or grazing on grassland biodiversity in recent years. Hence, you have to complete the discussion with the results that are now lacking, giving more detailed explanations and interpretations and comparing your results with previous syntheses results, and probably mentioned them in introduction section.

That's a good suggestion, thank you. We've added some additional citations to the paper Discussion to address this: Cao et al. 2024, Li et al. 2022, and You et al. 2017. Furthermore, in lines 322-327 of the Discussion, we compare our findings to past research.

We gained valuable new insights from these papers. We have cited Cao et al. 2024 as a previous global synthesis of grazing effects on plant diversity in lines 280 and 324 of the Discussion as we compare our findings to the prior literature. Li et al. 2022 provides evidence to support our hypothesis that nitrogen enrichment will have stronger effects on forbs than phosphorus enrichment, so we have cited paper in line 261 of the Discussion. You et al. 2017 takes a meta-analysis approach to poses similar questions about the effects of nitrogen addition on grass vs forb responses, so we have cited this paper in lines 152 in the Introduction and in lines 249 and 288 when connecting our work to past syntheses in the Discussion.

Cao F, Li W, Jiang Y, Gan X, Zhao C, Ma J (2024) Effects of grazing on grassland biomass and biodiversity: A global synthesis. Field Crops Research, 306, 109204.

Li W, Gan X, Jiang Y, Cao F, Lü X-T, Ceulemans T, Zhao C (2022) Nitrogen effects on grassland biomass production and biodiversity are stronger than those of phosphorus. Environmental Pollution, 309, 119720.

You, Chengming, et al. (2017) Grass and forbs respond differently to nitrogen addition: a meta-analysis of global grassland ecosystems." Scientific Reports 7(1), 1563.

L358 treatments repeated;
Thanks, we corrected this typo.

L377 with repeated ;
Thanks we corrected this typo

Cao F, Li W, Jiang Y, Gan X, Zhao C, Ma J (2024) Effects of grazing on grassland biomass and biodiversity: A global synthesis. Field Crops Research, 306, 109204.

Li W, Gan X, Jiang Y, Cao F, Lü X-T, Ceulemans T, Zhao C (2022) Nitrogen effects on grassland biomass production and biodiversity are stronger than those of phosphorus. Environmental Pollution, 309, 119720.

Thank you, we've cited these two papers in the discussion.

Reviewer #2 (Remarks to the Author):

A very solid study from the global NutNet Experiment that gives valuable insights into the interactive effects of multiple global change drivers on vegetation i.e. eutrophication, land use change (e.g. grazing) and drought.

I have only minor comments regarding the clarity of presentation which I attach as in-line comments in the main manuscript file.

Thank you, we have aimed to address the comments given in the file by making the following changes described below.

Line 107-108 & Line 159-161: We added a justification of the second hypothesis.

Line 168-169: We rewrote this using ecological instead of technical language.

Line 181: Simplified language as per reviewer 2's suggestion.

Line 183: We added percent change.

Lines 182-217: Made line edits per suggestions.

Line 235: Corrected typo.

Line 236-238: Added concluding sentence about role of herbivory.

Line 247: This is the case. Forbs typically contribute strongly to diversity in grasslands. Grass richness also concurrently declines with forb richness, suggesting communities are becoming dominated by a few grass species.

Line 254-256: Added sentence about the implications of forb losses for ecosystem services.

Line 261-267: Good idea, we added an additional line putting this in a global context.

Line 268-269: We added a sentence addressing the direction of the effect.

Line 277-279: We added a sentence about grasses.

Line 288-289: We've added a sentence about different aridity measures.

Line 381: We've introduced LRR as a term.

Methods: We corrected grammatical issues.

Thank you for the valuable feedback. We have updated the manuscript to address all of the reviewer comments below.

Reviewers' comments:

Reviewer #1 (Remarks to the Author):

This resubmitted manuscript has been well improved after revision.

Reviewer #2 (Remarks to the Author):

Dear authors,

I enjoyed reading the revised version of your manuscript and am overall satisfied with your amendments. A valuable publication with insightful results, which I will share with my students for sure, congratulations!

I was left with 2 tiny comments and one statistical question:

Comments:

L353: herbivory offsets the negative effects of fertilization

We corrected the grammar.

L360: ... as some forbs are ... (space was missing between some and forbs)

We added a space here.

Question:

The model description is very transparent and I agree with all measures you undertook for handling spatial, sampling-design-related and temporal dependencies in the data.

The only aspect I was missing was a note on which Anova type you used for testing the significances of effects maybe that is worth adding. Given that you used interaction terms, you probably used type I or better type III Anova. I would use type III as this will account for random effects and the temporal AR structure when testing the effect of any fixed effect (`car::Anova(..., type="III")`). And just to confirm, as I have been dealing with this lately, you retrieved the p-values from the Anova object in R as this uses the appropriate/chooses test for your data set (random effects etc). The summary object gives the estimates, but uses only t-tests to estimate the sign. Difference of estimates compared to 0 and with interactive categorical factors only shows comparisons to one factor level. See recommendation by Zuur et al. "Introduction to Mixed-Effects Models".

Thank you for bringing this to our attention; we agree with this point. We ran type III Anovas on our main set of models, using the car package in R. We have added this update to lines 412-414 of the Methods: "We further calculated model significance using a type III Anova to account for random effects and autocorrelation structures using the "car" package in R (see Supplemental Tables S11-S14)⁸⁸". We report our type III Anova results in Supplemental Tables 11, 12, 13, and 14. The results of using the Type III Anovas were the same as that of

using t-tests via the summary function, so this information did not change our interpretation of our results.

Best wishes

Lena Neuenkamp

REVIEWERS' COMMENTS:

Reviewer #2 (Remarks to the Author):

Dear authors,

thank you for clarification on the significance testing, with which I fully agree.

I have no further comments. Congratulations to this very insightful study!

Best wishes

Lena Neuenkamp

Thank you for the kind words and for taking the time to provide such thoughtful feedback throughout the submission process.